

# Joint analysis of coastal altimetry and HF radar data: observability of seasonal and mesoscale ocean dynamics in the Bay of Biscay

Ivan Manso-Narvarte[1], Ainhoa Caballero[1], Anna Rubio[1], Claire Dufau[2], Florence Birol[3]

[1]AZTI-Marine Research, Herrera Kaia, Portualdea z/g, 20110 Pasaia, Spain
[2] CLS Space Oceanography Division, 8-10 Rue Hermès, 31526 Ramonville St. Agne, France
[3] LEGOS, 14 Ave. E. Belin, 31400 Toulouse, France

*Correspondence to*: Ivan Manso (imanso@azti.es)

**Abstract.** Coastal HF radar systems provide operational measurements of coastal surface currents with high spatial (~1-5 km) and temporal (~hourly) sampling resolution while the near continuous altimetry missions afford, from 1993 to nowadays, information of geostrophic currents in the global ocean with typical along-track and temporal sampling resolutions of >7 km and >9 days, respectively. During the last years, the altimetry community has made a step forward in improving these data in the coastal area, where the data present lower quality than in the open ocean. The combination of HF radar and altimetry measurements arises as a promising strategy to improve the continuous monitoring of the coastal area (e.g. by expanding the measurements made by HF radars to adjacent areas covered by the altimetry, or by validating/confirming improvements brought by specific coastal algorithms or new altimeter missions). A first step towards this combination is the comparison of both datasets in the overlapping areas.

In this study, a HF radar system and two Jason-2 satellite altimetry products with different processing (CTOH and CMEMS) are compared over the period from 1 January 2009 to 24 July 2015. The results provide an evaluation of the performance of different coastal altimetry datasets within the study area and a better understanding of the ocean variability contained in the HF radar and both altimetry data sets. The variability of the radar and altimetry measurements is higher near the coast, and both observing systems detect the Iberian Poleward Current and eddies, which are the main mesoscale processes within the study area. The highest correlations between radar and altimetry (up to 0.64) take place in the slope, where the Iberian Poleward Current represents a significant part of the mesoscale variability. Besides, the use of a simple Ekman model to add the wind-induced current component to the to the altimetry-derived geostrophic currents increases the agreement between both data sets (increasing the correlation in around 10 %).

## 1 Introduction

Ocean dynamics result from a combination of processes of different time and space scales. However, and mainly due to technical limitations, this complexity cannot be captured by the existing observational systems if they are analysed individually since each observing technique is designed for resolving certain scales. Nowadays, there is a growing tendency to combine different observing systems for a more complete description and understanding of the ocean dynamics. Current observatories



are designed to monitor, in an operational way, the ocean environment to support the human activities concentrated in the coast. In the last years, a great effort has been focused on the development and improvement of these platforms. European projects such as JERICO and JERICO-NEXT have worked on the one hand, on the improvement and creation of coastal observatory networks, and on the other hand, on the applications of these observatories for addressing European marine policy.

As an example, the latter project aims to integrate the emerging methodology and technology, in order to proceed towards the automated monitoring at high temporal and spatial resolution of wider areas, to provide the best possible quality indicators for the European Marine Strategy Framework Directive. For this purpose, there is an ongoing research on assessing the interconnection among physics, biogeochemistry and biology, at different spatial and temporal scales. In addition to the development of coastal observatories, there are global initiatives, as the GLOBCURRENT project (2014-2017;

http://www.globcurrent.org/), aimed to advance in the evaluation of the synergy of satellite sensors and in situ data for the quantitative estimation of ocean surface currents (e.g. Rio et al., 2014*).

Among the different methodologies to retrieve surface currents, two of them are particularly interesting due to their high potential complementarity: satellite altimetry and HF radar (HFR) systems. The former technique consists in a constellation of altimeters onboard satellites measuring the global sea level with a revisit period higher than a week and a track distance

around tens of kilometres. These continuous sea level series are today close to complete 25 years of data, resolving the ocean dynamics from mesoscale to near climate scale. HFRs are designed to measure the local ocean surface dynamics with a high time and space resolution. However, altimetry and HFR do not capture exactly the same dynamics. Altimetry detects surface currents that are in geostrophic equilibrium (excluding the direct response of the surface layer to the wind and then part of the HF variations), whereas HFRs measure surface total currents, i.e. geostrophic and ageostrophic components (like wind driven

currents, inertial waves, or the wave-induced Stokes drift, e.g. Graber et al.,1997; Law, 2001; Ardhuin et al*., 2009).

Besides the effort made for collecting data from different platforms, methods for combining these data are under development. Recent studies focused on the evaluation of the capabilities of altimetry using HFR, concluded that HFR offers a way to improve the validation of altimetry products for coastal areas (Chavanne and Klein, 2010; Liu et al., 2012; Pascual et al., 2015; Troupin et al., 2015; Roestler et al., 2013). One of the most extended approaches found in the literature to study the synergy

between altimetry and HFR data consists on the comparison of the total across-track currents in the along-track direction (e.g. Morrow et al., 2017; Troupin et al., 2015; Pascual et al., 2015). The combination of HFR and altimetry could help to potentiate their strengths by for example, expanding the spatial and temporal coverage of the HFR systems or evaluating and correcting of the altimetric signal near coast.

In this study, we focus on the south-eastern Bay of Biscay (SE-BoB), which is characterized by abrupt changes in the

30 orientation of the coast, by a narrow shelf and slope and the presence of canyons (e.g. Capbreton Canyon). The characteristic winter surface circulation in the SE-BoB is mainly related to the Iberian Poleward Current (IPC), which affects the upper 300 m of the water column. In winter, the IPC flows over the slope, advecting warm surface waters (Le Cann and Serpete, 2009; Charria et al., 2013) eastwards along the Spanish coast and northwards along the French coast (Fig. 1). In summer, the flow is reversed being three times weaker than in winter (Solabarrieta et al., 2014). Overlaid to the density-driven slope circulation,



wind-induced currents are the main drivers of the circulation at the surface in the area (e.g. Lazure, 1997, Solabarrieta et al., 2015). During autumn and winter, SW winds dominate and generate northward and eastward drift over the shelf. The wind regime changes to the NE during spring, when it causes sea currents toward the W-SW along the Spanish coast. The summer situation is similar to that of spring, but the weakness of the winds and the greater variability of the direction of the general drift make currents more variable (González et al., 2004; Lazure, 1997; Solabarrieta et al., 2015). In addition to these processes, mesoscale eddies in the SE-BoB are generated, mainly during winter, by the interaction of the IPC with the abrupt bathymetry (Pingree and Le Cann, 1992) (Fig. 1). The combination of these processes makes the SE-BoB an area of interesting complexity. The existence of a long historical time series of surface currents from a long-range HFR system in the area provides, in turn, an invaluable opportunity to explore the benefit of a combined analysis of satellite and land-based remote sensing ocean currents.

The coastal observatory of the SE-BoB is composed by a network of oceano-meteorological coastal stations, two slope buoys and a HFR (two sites, see Fig. 1). The performance of this system and its potential for the study of the ocean processes and of transport patterns in the area have already been demonstrated by previous works (e.g. Solabarrieta et al., 2015; Rubio et al., 2018). With regard to the altimetry, several studies have been undertaken to study regional processes that goes from mesoscale (Dussurget et al., 2011; Herbert et al., 2011; Caballero et al., 2008b, 2014, 2016) to climate scale (e.g. Pingree, 2005).

The main objectives of this study are first, to obtain a diagnosis of the agreement of the surface currents measured by both methodologies over the SE-BoB and secondly to evaluate the observability of certain mesoscale processes by the altimetry and HFR.

## 2 Data and methods

### 2.1 Data

#### 2.1.1 HFR data

HFRs are remote sensing instruments that send radio waves to the surface ocean and use the signal backscattered by the surface ocean waves to measure the radial velocity of the surface current (toward or away from each HFR antenna). They can measure ocean surface currents over wide areas (reaching distances from the coast over 100 km) with high spatial (1-5 km) and temporal ($\leq 1$ h) resolution. In this study, surface currents were obtained by means of two long-range HFR antennas, owned by the Directorate of Emergency Attention and Meteorology of the Basque Government Security Department. The SE-BoB HFR antennas emit at a central frequency of 4.5 MHz and a 40-kHz bandwidth. They are located at Matxitxako and Higer Capes (Fig. 1) and provide operational data since 2009 (with some interruptions mostly due to maintenance stops or malfunctioning related with severe atmospheric conditions). The averaged Doppler backscatter spectrum obtained from the received signal (in a window of 3 h) is processed to obtain hourly radial currents using the MUSIC algorithm (Schmidt, 1986). The coverage of radial data is up to 150 km with a 5 km range cell resolution and 5° angular resolution. Radial data are quality controlled using



advanced procedures based on velocity and variance thresholds, noise to signal ratios and radial total coverage. Since the deployment of the HFR system, the receipt antenna pattern of the two HFR sites has been calibrated at least biannually. A more detailed description of the system, and of the HFR data validation exercises in the study area are provided by Solabarrieta et al. (2014, 2015 and 2016) and Rubio et al. (2011 and 2018).

To obtain total currents gridded into a 5 km resolution regular orthogonal mesh, a least mean square algorithm (spatial interpolation radius of 10 km) was applied. Then, using the same grid, radial velocities were processed with HFR_Progs to generate spatially gap-filled Open Mode Analysis (OMA) total currents (Kaplan and Lekien, 2007). The OMA was applied by using the HFR_Progs Matlab package (https://cencalarchive.org/~cocmpmb/COCMPwiki/), based on Gurgel (1994) and Lipa and Barrick (1983). 85 OMA modes, built setting a minimum spatial scale of 20 km, were used to generate hourly total fields.

A first analysis of the comparisons between HFR and altimetry showed that the results obtained using total currents generated by least mean square and OMA were very similar. Thus, only results using OMA currents are presented in this work.

The typical spatial scales resolved by the HFRs mainly depend on the resolution of the data; and thus, mainly on the operation frequency of the systems (Rubio et al., 2017). For the SE-BoB, the spatial scales resolved are typically of O(15-20) km.

### 2.1.2 Altimetry data

The basic principle of the altimetry technology is to send a radar signal to the sea surface and then to measure the return echo reflected by the sea surface. The time needed for the signal to go and come back determines the distance between the altimeter and the sea surface (called the range). A physically-based model (Brown, 1977) is adjusted to the resulting signal, called waveform, providing different parameters, among which the range. In order to reduce the measurement noise, the result is averaged, and the final data rate is classically (and in our case) of 1 Hz (i.e. one data every ~7 km along the satellite track).

The sea surface height (SSH) is obtained by subtracting the range to the satellite orbit altitude (with respect to the ellipsoid reference level) and by applying several corrections (tropospheric and ionospheric effects on the radar wave, sea surface bias…): SSH = orbit - range - corrections. To retrieve the total geostrophic currents, in balance with the SSH gradients, the surface height has to be referenced to the geoid: it is called Dynamic Topography (DT). However, since geoids are not known with enough accuracy, only geostrophic anomaly currents can be accurately derived from Sea Level Anomaly data, which are

referenced to a mean (Le Traon et al., 2003) by subtracting a mean profile to the SSH: SLA = SSH - <SSH>. For more information about the SLA and the reference surfaces, the reader is referred to https://www.aviso.altimetry.fr/en/techniques/altimetry/principle/basic-principle.html.

The two different along-track SLA time series used herein come from Jason-2 altimeter's 248 and 213 tracks from cycle 18 to 259 and have a period of ~10 days. One of the altimetry data sets used in this study is the CTOH-XTRACK product

(http://ctoh.legos.obs-mip.fr/products/coastal-products/coastal-products-1/sla-1hz), which provides SLA data specifically processed for coastal areas. The filtered version of the product was used, meaning that a 40 km cut-off spatial Loess filter was also applied in order to reduce the remaining noise in the along-track SLA.



For the same Jason-2 altimeter measurements, a differently processed data set was also used in order to assess possible discrepancies between the two altimetric products. This product is a preliminary version of the 2018 update of the reprocessed global ocean along-track L3 product provided by the Copernicus Marine Environment Monitoring Service (CMEMS). As for the CTOH-XTRACK product, the filtered version was used with a 65 km cut-off spatial Lanczos filter in order to reduce noise (Pujol et al., 2016).

Note that track 248 covers the HFR footprint area, whereas track 213 only crosses a small area at the NW of the HFR total currents coverage (Fig. 1); therefore, track 213 is only useful for the comparison with HFR radial data.

### 2.1.3 Wind data and SST images

Hourly wind data from the Weather Research and Forecasting model (WRF, http://mandeo.meteogalicia.es/thredds/catalogos/WRF_2D/catalog.html) were provided by the meteorological agency of Galicia (MeteoGalicia). This model, with a native resolution of 12 km, reproduces with a reasonable accuracy the offshore wind fields of the SE-BoB (Ferrer et al., 2010). In this study, the WRF gridded field was interpolated to the Jason-2 along-track points.

Due to the cloudy weather in the SE-BoB, the most proper infrared SST images were selected one by one from the AVHRR sensor series of 1 km resolution to process Level 2 SST maps.

### 2.2 Methods

Since the time resolution of the altimetry (>7 d) is lower than that of the HFR (hourly), the HFR currents data were filtered using a low-pass filter, based on a 10-day running average. The objective was to remove the HF signals contained in the data (see for instance Solabarrieta et al., 2014) and part of the ageostrophic signals to make HFR data closer to the measurements from altimetry data. Several tests were carried out to determine the sensibility of the results to the temporal window chosen for the filter. For that purpose, 2, 5, 10 and 15 d low-pass filters were tested, and the resulting currents were compared to those derived from the SLA. The 10-d filtered HFR currents provided the highest correlation with the altimetry data. It is worth noting that the 10-d running average filtered out a significant part of the wind-induced currents, except the low-frequency (LF) Ekman component.

Since from the altimetry data used here we can only obtain sea surface anomaly currents, as explained in Sect. 2.1.2, the comparisons with the radar data were carried out in terms of anomaly. In order to obtain the HFR anomaly currents, the temporal average of the HFR currents for the study period was subtracted to the series of low-pass HFR currents (Eq. (1)). Then, to obtain SLA relative to the same period, the average SLA for the study period was subtracted to the SLA series (Eq. (2)).

$$AC_{HFR} = V - \overline{V_{t1-t2}} \tag{1}$$

$$SLA' = SLA - \overline{SLA_{t1-t2}} \tag{2}$$



Hence, the time referenced SLA′ and $AC_{HFR}$ anomalies were obtained as suggested in Pujol et al. (2016), where $t1$ = 1 January 2009 and $t2$ = 24 July 2015. Hereafter, when referring to HFR current anomalies computed using Eq. (1), we will use $AC_{HFR,R}$ to refer to anomalies computed directly from the radial HFR components and $AC_{HFR,T}$ to refer to anomalies computed from total OMA HFR currents.

For the altimetry velocity, across-track geostrophic currents were inferred by means of the finite difference geostrophic velocity equation (Eq. (3)),

$$AC_G = -\frac{g}{f}\frac{\Delta\text{SLA}'}{\Delta x}$$ (3)

where $g$ is the acceleration of gravity, $f$ is the Coriolis parameter, $x$ is the along-track distance, SLA′ is the time referenced SLA and $AC_G$ is the geostrophic current anomaly. $AC_G$ was estimated along the altimeter track by a 3 point central difference operator.

For the statistical comparison between $AC_G$ and $AC_{HFR}$, two different strategies were used. The first and simplest approach is the comparison of HFR radial across-track currents with altimetry across-track currents in two given points (Sect. 2.2.1), which permits the direct use of radial HFR currents. The second approach is the along-track comparison (Sect. 2.2.2) of total HFR and altimetry across-track currents, which provides additional information on the spatial variability of the agreement between both data sets. Finally, to take into account the LF Ekman component that remains in the low-pass filtered HFR data, a simple model for the computation of this component was used (Sect. 2.2.3).

### 2.2.1 Pointwise comparison

This method, previously used by Liu et al. (2012), consists in a direct comparison between HFR and altimetry data in a certain point where one of the HFR radial directions (red lines in Fig. 2) crosses the altimeter track perpendicularly. This approximation allows to directly use the radar radial currents, which are in the same direction than the across-track $AC_G$. This method was applied for Jason-2 satellite's 213 and 248 tracks shown in Fig. 1, by using the corresponding $AC_{HFR,R}$ measured by the Matxitxako antenna in two points: point E on the track 248 and point W on the track 213 (Fig. 2). Note that the radials from Higer station were not used since they did not provide any orthogonal radial direction for track 248 and were too gappy for the track 213 due to the large distance between this track and the antenna. Point E is located near the Capbreton Canyon at 43.75º N, 2.05º W, 46.86 km far from the coast and in a depth of 500 m (on the slope); whereas, point W is located at 43.80º N, 3.58º W, 40.73 km far from the coast and in a depth of 3000 m (on the abyssal plain). It is worth noting that using directly radial currents, additional errors that propagate in the combination from HFR radials into HFR OMA currents are avoided.

In order to make the computations more robust to the potential absence of HFR or altimetry data in the points W and E, nearby points were considered to obtain the across-track currents (Fig. 2). With regard to $AC_{HFR,R}$, radial directions of $\pm$ 5º (blue lines), away from the orthogonal radial direction (red line) were also considered. For each of them, the points in a range of $\pm$ 5 km from the track (blue crosses for the adjacent radials and red crosses for the central radial) were selected. Then, the across-track





$AC_{HFR,R}$ in our point was obtained by firstly averaging the values for each radial, so that only three values along the track were obtained (green crosses). Finally, the three values were again averaged to get the corresponding across-track $AC_{HFR,R}$ (black dot).

Then, the $AC_G$ in the three along-track points considered for the HFR case (green crosses) were averaged to obtain the $AC_G$ in the central point (E and W). This permitted to ensure a similar spatial smoothing for both data sets. It must be mentioned that at point W and for CTOH altimetry product, the points were located between the along-track HFR points, so instead of considering three along-track points, two points on each side of the central point (W) were selected.

### 2.2.2 Along-track current comparison

In order to assess the variability of the comparison between HFR and altimetry from the coast to the open ocean, the comparison between the across-track $AC_{HFR,T}$ and $AC_G$ along the 248 track was performed. For that purpose, the $AC_{HFR,T}$ currents were interpolated into the along-track altimetry points, and they were rotated to the across-track direction. $AC_G$ were obtained by means of the finite difference geostrophic velocity equation (Eq. (3)) and by means of a 3 point central difference operator, like in the pointwise comparison. Then, an average with its adjacent two points on each side was carried out. As in the previous case, this permitted to ensure a similar spatial smoothing for both data sets. The sensitivity to the number of adjacent points considered was tested, and this approach was the one that provided the best adjustment to the HFR data.

### 2.2.3 Ekman currents

Ekman currents were estimated to find out which was their contribution to LF currents in the area and to evaluate how this component contributed to part of the differences observed between HFR and altimetry. Three different ways to calculate Ekman currents were tested to infer which one provided the best results in the comparisons: the rule of thumb that states that the surface currents are 3 % of the wind velocity, Ekman equations for the surface (Ekman, 1905) and the model M1 proposed in Rio and Hernandez (2003). Finally, the M1 model offered the best results:

$$\vec{u}_{ek} = \frac{b\vec{\tau}}{\sqrt{f}} e^{i\theta} \tag{4}$$

where $f$ is the Coriolis parameter, $\vec{u}_{ek}$ is the Ekman currents vector (meridional and zonal components), $b$ is the amplitude parameter, $\theta$ is the phase parameter and $\vec{\tau}$ is the wind stress vector (meridional and zonal components) that was obtained by means of:

$$\vec{\tau} = \rho_{air} C_d |\vec{w}| \vec{w} \tag{5}$$

where $\rho_{air}$ is the density of the air (1.22 kg m⁻³), $\vec{w}$ is the wind vector and $C_d$ the drag coefficient. $b$ and $\theta$ were acquired adjusting the model (by a least square fit) to the real data in the study area. These parameter values were taken from Caballero et al. (2008a), where from the fit of wind stress measurements and drifter-derived currents of the Bay of Biscay to Eq. (4) the




following results were obtained: $b$=4.45·10$^{-3}$ m$^2$ kg$^{-1}$ s$^{1/2}$ and $\theta$=-23.68°. Once the parameters were obtained, $\vec{u}_{ek}$ was estimated using the wind data series described in Sect. 2.1.3.

Ekman currents initially computed in the locations of the WFR model nodes were interpolated and rotated (from zonal and meridional directions to along-track and across-track directions). For the pointwise comparison, they were interpolated in E, W and in their adjacent points, and rotated to obtain the across-track component. After, they were averaged so that finally, the across-track Ekman current velocity was obtained in each point (E and W). In the along-track current comparison, they were interpolated to the altimetry along-track points and after rotated to get the across-track component. Ekman currents were also 10-d running averaged to remove the HF Ekman signal.

For the comparisons with $AC_{HFR}$, Ekman anomaly currents ($AC_E$) were obtained by subtracting the average value of the study period like in Eq. (1) and (2), and, were added to $AC_G$; this is indicated as $AC_G+AC_E$ on Table 1.

The results of all the comparisons described above are presented in terms of correlation ($r$) and the root mean square difference (RMSD), and the mean and the standard deviation ($std$) of the across-track currents measured by each system. All these parameters were computed for the study period. Since the HFR radials are used in the pointwise comparison, the crossing points are called E$_R$ and W$_R$. In the along-track comparison, the across-track currents are compared along all the track. However, one of those along-track points is the point E and considering that the corresponding statistical parameters are the ones that use the rotated $AC_{HFR,T}$, the point is named E$_T$. The Max$_T$ points are the points where $r$ is maximum in the along-track current comparison (point 2.10° W, 43.82° N for CTOH and point 2.09° W, 43.80° N for CMEMS); the rotated $AC_{HFR,T}$ were also used in these points. All these points and their statistics are displayed in Table 1, as well as in Figs. 3, 4 and 5, and they are also discussed in the next section. Moreover, in Fig. 5, apart from the results of E$_T$ and Max$_T$, the results for all the track 248 are also shown.

Note that the HFR-altimetry comparisons were carried out for CMEMS and CTOH data sets and that each comparison was also made with and without adding $AC_E$.

## 3 Results and discussion

### 3.1 Statistical results

Despite the differences between the two measuring systems, the comparison between $AC_{HFR}$ and $AC_G$ ($AC_G + AC_E$) shows maximum correlations of 0.64 (0.70). Table 1 provides an overview of the statistical results of this comparison in the points E$_R$ and W$_R$ for the pointwise method and in the points E$_T$ and Max$_T$ for the along-track method.

Concerning the pointwise comparisons between $AC_G$ and $AC_{HFR,R}$ it can be observed that $r$ depends on the location, as well as on the SLA product considered. In any case, the addition of $AC_E$ increases $r$ by 6-11 % for both SLA products; from 0.53 and 0.48 to 0.64 and 0.59 in E$_R$, and from 0.61 and 0.64 to 0.67 and 0.70 in W$_R$, for CMEMS and CTOH, respectively. The RMSD also decreases between 0.19-0.41 cm s$^{-1}$. Therefore, in general terms, the addition of $AC_E$ decreases the differences between





$AC_{HFR,R}$ and $AC_G$. However, it adds variability; the *std* for $AC_G + AC_E$ increases 0.69-1.01 cm s$^{-1}$. It should be pointed out that the *std* of $AC_{HFR,R}$ is lower than the *std* of $AC_G$ (where CMEMS's variability is higher than CTOH's), probably because HF signals may have been better filtered for the radar due to the 10-day running average temporal filter or because the finite difference operator that estimates the geostrophic velocity from SLA′ increases it. The same effect was observed in Liu et al.
(2012) and in Verron et al. (2018). However, although it increases even more the variability the addition of $AC_E$ clearly improves the results.

Fig. 3 displays the time series of $AC_{HFR,R}$ and $AC_G + AC_E$ in the points E$_R$ and W$_R$. In general, there is a global agreement between the three series ($AC_G$ (CTOH) $+ AC_E$, $AC_G$ (CMEMS) $+ AC_E$, $AC_{HFR,R}$), and all the data sets are observing the same variability. Although in Fig. 3 it is difficult to observe any differences between locations and data sets, the lowest RMSD is
observed in E$_R$ for CMEMS $AC_G$ (RMSD= 7.14 cm s$^{-1}$), while RMSD rises up to 8.95 cm s$^{-1}$ in W$_R$ for CTOH. A larger variability in W$_R$ is observed in Table 1, and it is also noticeable in Figs. 3a and 3c.

In terms of correlation, the results suggest a higher agreement in W$_R$ (0.67 and 0.70 for CMEMS and CTOH respectively) than in E$_R$ (0.64 and 0.59 for CMEMS and CTOH respectively). This could be explained by the fact that point W$_R$ is located in a deeper area, more influenced by the IPC that flows over the slope. The IPC shows lower variability in point W$_R$ than in point
E$_R$, where the signal of the slope current is less persistent. Indeed, point E$_R$ is located near the Capbreton Canyon, an abrupt bathymetry area where the coast also changes of orientation. These spatial differences agree to what was observed in the area by Rubio et al. (2009), from the analysis of the ADCP time series of two buoys located over the slope in locations comparable to points E and W. These authors suggest that while there is a clear along-slope transport with intense mesoscale variability at Matxitxako buoy (slightly east from point W), at Donostia buoy (close to point E) the influence of the slope circulation was
less significant and stronger stratification was observed. The lower current velocities and lower vertical coherence observed at Donostia buoy during winter could be linked to the complex bathymetry, which might force the IPC to flow over deeper grounds in the area. This could also explain why the addition of $AC_E$ increases more *r* in E$_R$ (by 11 %) than in W$_R$ (by 5-6 %), where the circulation has a stronger geostrophic component.

When the performances of the two SLA products are compared, it must be highlighted that CTOH shows higher (lower) *r*
(RMSD) in W$_R$, while CMEMS shows higher (lower) *r* (RMSD) in E$_R$. However, the differences are small and do not permit to discuss on the relative accuracy of the two SLA products.

In E$_R$ (W$_R$) when *r* is bigger for CMEMS (CTOH) than for CTOH (CMEMS), the RMSD is lower in the former. Therefore, the higher is the correlation, the smaller is the difference between $AC_G$ and $AC_{HFR,R}$. However, the difference of the RMSD in each point (i.e. between the RMSD value of each altimetry product with the radar) is not related to the difference in *r*.
Figs. 3b and 3d also show the residuals between $AC_G$ and $AC_{HFR,R}$ for each altimetry product. It can be observed that in W$_R$ (Fig. 3d) the residuals' amplitudes are larger for CMEMS than for CTOH, agreeing with the higher RMSD value observed for the CMEMS $AC_G$. In E$_R$ the similar RMSD values observed for both SLA products are in agreement with the similar amplitudes of the residuals in Fig. 3b.





The IPC winter intensification is visible in all data sets, being stronger in $AC_G$ for both SLA products. There are some remarkable intensifications, for instance in November 2014, where a strong peak is shown in all the series and where the HFR current signal is higher in $W_R$ than in $E_R$ (see black arrows in Figs. 3a and 3c), approaching more to the $AC_G$. There is another remarkable intensification in winter 2009, when the IPC shows a pronounced peak (see black arrows in Figs. 3a and 3c),

especially in $E_R$, where CTOH is very similar to HFR. The intensification in point $W_R$ is not so clear, but it is still noticeable comparing with the rest of the period. Equatorward slope current intensifications can be also observed through the whole period, as for instance in May 2011, where the peak is more prominent in $W_R$ than in $E_R$ (and even more for CMEMS) (see black arrow in Figs. 3a and 3c).

Since the presence of a stronger IPC signal is expected to improve the correlation between HFR and SLA data sets, and the

10 IPC shows marked seasonality, a monthly analysis has been carried out (Fig. 4). Every monthly value of every statistical parameter shown in the figure, has been computed taking into account all the available data for that month during all the study period.

It can be observed that in terms of monthly mean currents, the three series have the same tendency and that in general there is low discrepancy among them. No significant differences in terms of monthly patterns are observed among the two SLA

products. The winter poleward current intensification is patent from October to January with a maximum in November (ranging for all datasets from 7.51 to 13.36 cm s$^{-1}$ in $E_R$ and from 11.73 to 14.79 cm s$^{-1}$ in $W_R$). In $E_R$, the current intensification is still perceptible in February, whereas in $W_R$ there is a little increase in March in the $AC_G$, but not in the $AC_{HFR,R}$ currents. From March to September the mean is maintained around zero, with a slight decrease reaching the minimum in September (ranging from -7.30 to -3.86 cm s$^{-1}$ in $E_R$ for all datasets and from -5.37 to -3.56 cm s$^{-1}$ in $W_R$) and showing an equatorward weak mean

slope current for that period. The addition of $AC_E$ slightly strengthens the intensity of the slope currents for both poleward and equatorward directions. This can be explained by the general wind patterns of the area, which are in agreement with the main geostrophic regime in the study area, although winter south-westerlies are stronger than the summer north-easterlies (Herbert et al., 2011).

Fig. 4 shows, as it has been mentioned previously, lower *std* values for the $AC_{HFR,R}$ along all the period, whereas the addition

of $AC_E$ increases it slightly, especially in winter when winds are stronger. In point $E_R$, the tendency of the *std* is similar to that of the mean, being slightly higher in January, and then stable (with small oscillations) until September. In the last three months of the year it is increased. Therefore, there is a higher variability in late autumn and winter, probably due to the slope current intensification and the effect of the wind (as mentioned before).

In point $W_R$, the highest variability takes place in the first 4 months of the year. This increase is coherent with the intensification

of the slope current and the development of an anticyclonic structure in March and especially in April near Torrelavega Canyon (Caballero et al., 2014). Afterwards, the variability is practically maintained, with little oscillations and an increase in CMEMS data.

Regarding the RMSD, the patterns are similar to those of the *std* of the $AC_G$ but with variability. In general, the addition of $AC_E$ improves the results by a slight decrease of the RMSD. The major differences between $AC_G$ and $AC_G + AC_E$ time series,





are observed in February and April in point $E_R$. This can be clearly seen in the normalized RMSD, which measures the value of the RMSD with respect of the values of the $AC_{HFR,R}$.

The same statistical parameters computed along the track 248 are shown in Fig. 5 in order to study the spatial variability in the comparison between $AC_{HFR,T}$ and $AC_G$. Temporal statistics considering all the study period for each point of the track are plotted as a function of the distance to the first point of the track.

The $AC_{HFR,T}$ shows mean values close to zero along all the track, with low variability. Currents are oriented poleward over the shelf and upper slope, over grounds shallower than 1000 m, (i.e. until the point where the track leaves the slope current area). From there on, the mean is oriented equatorward. The mean $AC_G$ are also close to zero, however, they show larger variability changing between positive and negative values along the track, and with a lack of agreement between both SLA products in some points. Nevertheless, it should be pointed out that in any case the mean anomaly currents are very weak. The addition of $AC_E$ does not cause any spatial variation and it barely changes the values.

With regard to the variability, it is higher close to the coast. For $AC_G$, it softly decreases as it gets away from the first point of the track, until the 1000 m isobath of the slope is reached (where the grey area ends in the figure). From there on, it is kept almost constant with a slight local maximum at around 120 km from the first point. For the $AC_{HFR,T}$, the variability decreases until the point $E_T$ is reached (where the Capbreton canyon area is left) and after, it is also almost constant with two maximums at around 60 km and 105 km from the first point.

It can be once again observed that the addition of the $AC_E$ slightly increases the variability (Fig. 5), that the variability of $AC_{HFR}$ is lower than the one of $AC_G$ and that CMEMS's variability is higher than CTOH's. Moreover, in Fig. 5, all these results are proved for all the track instead of for some certain points.

The highest $r$ is observed between 200 and 1000 m isobaths of the slope, where the IPC signal is the strongest, and thus the geostrophic component measured by the HFR is also stronger. In the points furthest from the coast the $r$ decreases. This can be linked to the absence of a strong and persistent geostrophic component and a higher noise to signal ratio for the HFR data (which increases as we get away from the antennas).

The maximum (minimum) values of $r$ (RMSD) take place at around 37 and 45 km from the first point of the track for CMEMS and CTOH, respectively (Max$_T$ points). These values could be explained by the fact that those points are located in the middle of the slope, where the slope current is stronger and where they are out of the Capbreton canyon area. At the same time, in that area, the slope current direction is nearly orthogonal to the track, so that the across-track component is stronger. For CTOH, the values around the maximum are relatively high, that is why the maximum is not a prominent peak. The same happens for CMEMS, but with a sharper peak and higher value (see Table 1). The addition of $AC_E$ increases the $r$ by 8-10 % as it can be seen in the table (for $E_T$ and Max$_T$ points).

In general, the addition of $AC_E$ slightly increases (decreases) the $r$ (RMSD), along all the track. This fact is not perceptible in the figure, where it seems that the addition of $AC_E$ does not make any difference, but it can be observed in the values of the Table 1, except in the point $W_T$ for CMEMS data.



### 3.2 Observability of mesoscale processes in HFR and altimetry data sets

In order to provide a complementary insight to the synergies and differences between HFR and altimeter data, in this section the observability of different processes detected by HFR and altimetry is qualitatively analysed. Since the data are spatially filtered (for $AC_G$) or time filtered (for $AC_{HFR}$ and $AC_E$), the detectable processes are mesoscale, seasonal and interannual

processes, such as mesoscale eddies and the IPC. Only CMEMS data is used for this analysis, since the statistical results are very similar for both SLA products and CMEMS data have less data gaps in the period and study area, which is more suitable for monitoring ocean processes.

### 3.2.1 Observability of the IPC

Along-track values of SLA′, $AC_E$ and across-track $AC_{HFR,T}$ for all the study period are shown in Fig. 6. The highest SLA values

are observed during late autumn and winter, whereas the lowest ones are observed in spring and summer, specially from March to July (Fig. 6a). This is coherent with the contribution of the main driving factor of the seasonal SLA variability in the area, the steric effect. It was observed in Caballero et al. (2008b) that the maximum amplitude of sea level is reached in October in the BoB, whilst the minimum takes place in April. $AC_E$ shows a poleward seasonality with intensifications mainly in autumn and winter (usually from November to February), and weaker equatorward currents in spring and summer (usually from March

to October) (Fig. 6b). This fact agrees with the general wind pattern in the area. Along-track SLA gradients indicate winter slope current intensifications (IPC) mostly from November to January and from the coast to the 1000 m isobath approximately. The poleward intensification in winter 2014/2015 is the most remarkable, which was already observed in Rubio et al. (2018). On the other hand, in spring and summer the gradients are weaker and even suggest equatorward currents along all the track. The IPC intensifications are also detected by the $AC_{HFR,T}$ in late autumn and winter, mainly in the nearest points to the coast

between the 200 and 1000 m isobaths (Fig. 6c). In spring and summer, although there are also some poleward current pulses, they are weaker. During this period, equatorward current pulses are also observed. From the joint analysis of SLA′ and $AC_{HFR,T}$ data series, four main IPC events can be detected along the study period (Figs. 6a and 6c).

More details on these events are provided in Fig. 7, where selected HFR total currents (obtained from OMA as explained in Sect. 2.1.1) snapshots are shown. Although each event is represented for a specific date, they last around 2-3 weeks (not

shown), being the dates displayed in the figure representative of all the period. Note that the SST maps do not show exactly the same date as HFR snapshots and SLA′ data, due to the limitations of this technique under cloudy conditions. For the four events, HFR total currents show a typical IPC spatial pattern, with poleward circulation along the slope intensified between the 200 and 1000 m isobaths (Solabarrieta et al., 2015). The poleward patterns observed by the HFR agree with the $AC_G$ observed along the altimetry track, which show poleward currents intensified over the slope. In the four events, the SST images

show intensifications along the slope that are related to an increase of 0.5-1 °C along the time (not shown), which is the increase in temperature that typically correspond to the slope current intensification in the study area (Esnaola et al., 2013). The spatial extension of the warm water masses and the IPC along the French shelf/slope depends on the event and determines the area




where it is observed the highest agreement between the $AC_G$ and HFR total currents. During the IPC event of November 2009, the warm water tongue is closer to the coast. In this event, the strongest agreement between $AC_G$ and HFR total currents is observed over the slope, while they disagree in the north-western area of the domain. Otherwise during December 2010, the warm water extends over adjacent offshore areas, and so it does the area presenting the strongest agreement between HFR and altimeter. In three of the four events, the $AC_G$ and HFR total currents show lower fit over the shelf. These observations corroborate the results obtained in Sect. 3.1, where the best statistical results are obtained for the data pairs inside the slope area (Fig. 5).

The relationship between the IPC and the NAO in the study area was described in Garcia-Soto et al. (2002, 2004). They concluded that for strong IPC years, January water warmings (as a signal of the IPC) were related with negative NAO index values in the previous months (November, December). On the other hand, the Eastern Atlantic (EA) is also considered another possible factor of the IPC intensification, with positive EA values related to current intensifications. For the four events studied here, the relationship between IPC intensification and those indexes are shown in Table 2. In general, the NAO (EA) indexes are negatively (positively) related to the IPC in strong intensification periods, however, this is not always true (see for instance the event on January 2011, where positive NAO and negative EA index are observed). Moreover, the intensity of the currents is not related to the amplitude of the index, reaching the same conclusion as Le Cann and Serpette (2009) and Le Hènaff et al. (2011).

### 3.2.2 Observability of mesoscale eddies

As in the previous section, Fig. 8 shows four examples of eddies detected by the HFR and the altimeter. Although the effect in terms of statistical results of the presence of mesoscale eddies has not been explored, there is a qualitative agreement between $AC_G$ and HFR total currents when eddies are observed in the area covered by the two measuring systems (even if the eddy core is not crossed by the altimeter track). This happens when either the eddies cross the track of the altimeter or when the size of the eddies is large enough to be observable by the altimetry. Across-track $AC_G$ are generally in agreement with the HFR current fields, mainly in terms of current directions. For example, on the 15 September 2012 the altimeter crosses a small anticyclone located in the French slope, to the north of the Capbreton canyon head. The maximum SLA' and minimum $AC_G$ near the core of the eddy and the patterns of the SLA' and $AC_G$ north and southwards of the core agree with the structure detected by the HFR. An eddy with a similar diameter and located near the same area, is observed on the 15 February 2015. In this case the eddy is cyclonic, and though the HFR and altimetry currents in the area occupied by the structure agree, this is not the case in the rest of the track. Also, during winter, on the 15 March 2014, a cyclone is detected by both datasets near the head of the Capbreton canyon, in addition to another cyclone in the north-western part of the domain. Finally, on the 7 December 2014, an anticyclone, more energetic than all the former eddies, is observed over the Capbreton canyon. This anticyclone was analysed by Rubio et al. (2018), showing that it had an important role in the offshore transport of coastal waters. North of this eddy, the altimetry and the HFR detect a cyclonic circulation, but in this case, it is not clear from the HFR total current fields that the structure is an eddy.



## 4 Summary and conclusions

In this study, we have investigated the synergies and differences between land-based HFR and satellite altimetry, two remote sensing techniques that provide measurements of the ocean surface currents at different temporal and spatial scales. A general agreement between HFR and altimetry has been observed in the study area, with correlations ranging up to 0.7. The

comparisons have been carried out in terms of time anomaly of currents, following different approaches with radial and total OMA HFR data, and in all cases the addition of the LF Ekman component ($AC_E$) to the geostrophic component ($AC_G$) has improved the results.

The best agreement between both data sets is observed in the slope area, mainly between 200 and 1000 m isobaths, where the surface circulation is dominated by a more energetic geostrophic component. In the coastal area, the agreement between both

data sets is lower, and $AC_G$ has higher variability than $AC_{HFR}$. This can be explained by error propagation in the finite difference operator that estimates the geostrophic velocity from SLA′ or by a more effective filtering of HF signals in the radar data (where HF signals are correctly sampled and can be efficiently removed using a running average temporal filter).

In terms of monthly mean currents, north-eastward currents are observed in all data sets in late autumn and in winter, while weaker north-eastward and south-westward currents are observed in spring and summer. In the winter period, higher variability

is also observed in points E and W, probably linked to a more energetic slope current regime. Additionally, high variability is observed in point W in March-April, probably linked to an anticyclonic structure near Torrelavega Canyon.

Four IPC events have been isolated and described further by means of additional SST data. From this analysis we conclude that during the IPC intensifications, the qualitative agreement between $AC_G$ and HFR total currents is high, and well related to the SST anomalies. Moreover, a relationship between strong IPC events with negative (positive) NAO (EA) indexes in the

previous months has been detected for three of the four events. Although the effect in terms of statistical results of the presence of mesoscale eddies has not been explored, there is a qualitative agreement between $AC_G$ and HFR currents when eddies are observed in the area.

The low correlation between HFR and altimetry observed in some areas and periods can be due to several factors. It must be kept in mind that both technologies are based on different physical approaches to observe the surface ocean currents, at

different spatial and temporal scales, and work under different physical assumptions. Besides, the quality of the radar data is expected to decrease in the furthest points from the antennas and varies in function of the angle formed by the radial current components used for total current estimations (affecting the along-track comparison). Altimetry has also its own limitations and might have errors in the data editing procedure or in the corrections.

Future work should be oriented to a better understanding of the relationship of the surface circulation and the dynamics of the

subsurface layers, through the combination of remote observations with data in the water column. In addition, further comparison with HFR data and higher resolution altimetry products would enable a better understanding of the differences between both observing systems.



## 5 Acknowledgments

This study has been supported by the JERICO-NEXT project, funded by the European Union's Horizon 2020 research and innovation programme under grant agreement No 654410. The work of A. Rubio, was partially supported by the LIFE-LEMA project (LIFE15 ENV/ES/000252) and the Directorate of Emergency Attention and Meteorology of the Basque Government. This study has been also undertaken with the financial support of the Department of Environment, Regional Planning, Agriculture and Fisheries of the Basque Government (Marco Program). I. Manso was supported by a PhD fellowship from also the Department of Environment, Regional Planning, Agriculture and Fisheries of the Basque Government. Altimetry data used in this study were developed, validated, and distributed partly by the CTOH/LEGOS, France and partly by Collecte Localisation Satellites (CLS), France as the Sea Level Thematic Assembly Center of the Copernicus Marine Environment Monitoring Service. The SST Level 2 images were produced and distributed by the NERC Earth Observation Data Acquisition and Analysis Service (NEODAAS, http://www.neodaas.ac.uk/data). This is contribution number XXX, of the Marine Research Division of AZTI-Tecnalia.

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





**Table 1.** Statistics of different points for the study period. $r$ is the correlation parameter and RMSD is the root mean square difference of the velocity anomalies between HFR and altimetry for different altimetry products (i.e. CMEMS and CTOH). The mean and the standard deviation (*std*) are also estimated for each data set. E and W are the points where HFR radial velocities are orthogonal to the tracks 248 and 213 respectively (see Fig. 1). $E_T$ means that the parameters are estimated in the point E but instead of using the HFR radial velocity anomalies, using the HFR current velocity anomaly fields rotated to across track direction. That HFR data has also been used for $Max_T$ which is the track point where the correlation r is maximum (point 2.10º W, 43.82º N for CTOH and point 2.09º W, 43.80º N for CMEMS). RAD refers to HFR data, which really means radial anomaly current for $E_R$ and $W_R$, and across-track rotated OMA anomaly currents for $E_T$ and $Max_T$.

| | | \(r\) | | RMSD (cm s⁻¹) | | mean $\pm$ *std* (cm s⁻¹) | | |
|---|---|---|---|---|---|---|---|---|
| | | CMEMS | CTOH | CMEMS | CTOH | CMEMS | CTOH | RAD |
| $E_R$ | $AC_G$ | 0.53 | 0.48 | 7.37 | 7.55 | 0.35±8.66 | -0.06±8.53 | 0.11±5.94 |
| | $AC_G + AC_E$ | 0.64 | 0.59 | 7.14 | 7.36 | 0.35±9.67 | -0.21±9.38 | |
| $W_R$ | $AC_G$ | 0.61 | 0.64 | 9.24 | 7.83 | -0.17±11.90 | 0.25±9.88 | 0.03±9.30 |
| | $AC_G + AC_E$ | 0.67 | 0.70 | 8.95 | 7.42 | -0.18±12.89 | 0.19±10.57 | |
| $E_T$ | $AC_G$ | 0.56 | 0.53 | 7.15 | 6.51 | -0.11±8.25 | -0.30±7.59 | 0.16±5.45 |
| | $AC_G + AC_E$ | 0.65 | 0.62 | 7.15 | 6.43 | -0.12±9.31 | -0.42±8.56 | |
| $Max_T$ | $AC_G$ | 0.60 | 0.55 | 6.54 | 6.11 | -0.25±7.87 | -0.19±7.25 | - |
| | $AC_G + AC_E$ | 0.68 | 0.65 | 6.60 | 5.97 | -0.26±8.95 | -0.30±8.27 | |





**Table 2.** NAO and EA indexes in the previous two months of the events. m-2 means two months before the event whereas the m-1 means one month before.

| Event | NAO | | EA | |
|---|---|---|---|---|
| | m-2 | m-1 | m-2 | m-1 |
| November 2009 | 1.62 | -0.61 | 0.59 | 0.96 |
| January 2011 | -1.84 | -1.8 | 0.24 | -0.49 |
| January 2014 | 0.81 | 0.79 | 0.09 | 1.3 |
| November 2014 | 1.72 | -0.87 | 0.2 | 1.02 |



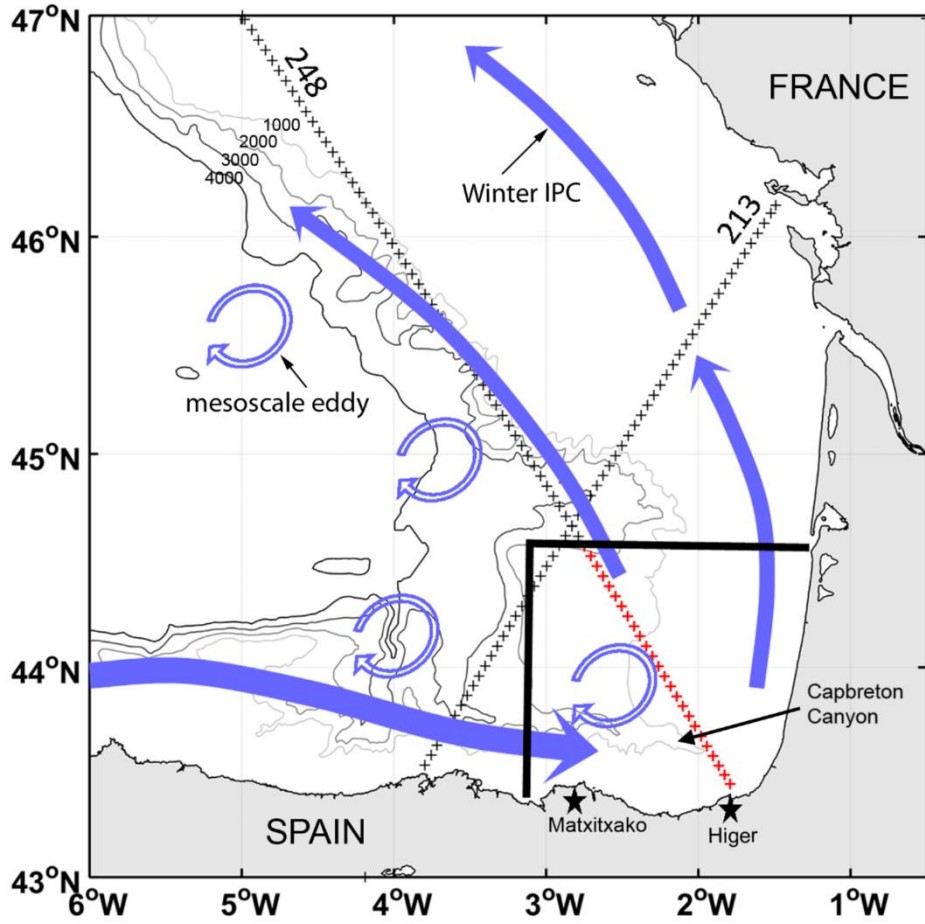

**Figure 1**. Study area, observational systems and main characteristics of the ocean circulation (Figure modified from Rubio et al. (2018)).
The winter IPC is represented by blue solid arrows, whereas the blue hollow arrows show the mesoscale eddy regime (although only
anticyclonic arrows are represented, eddies of anticyclonic and cyclonic polarity are observed in different locations along the slope). The
bold black lines delimit the HFR total currents footprint. The black stars represent the HFR stations: Matxitxako (left) and Higer (right).
Jason-2 tracks 213 and 248 are represented by black crosses and the part of the track used in this study is marked in red. Grey lines: 1000,
3000, and 4000 m isobaths.





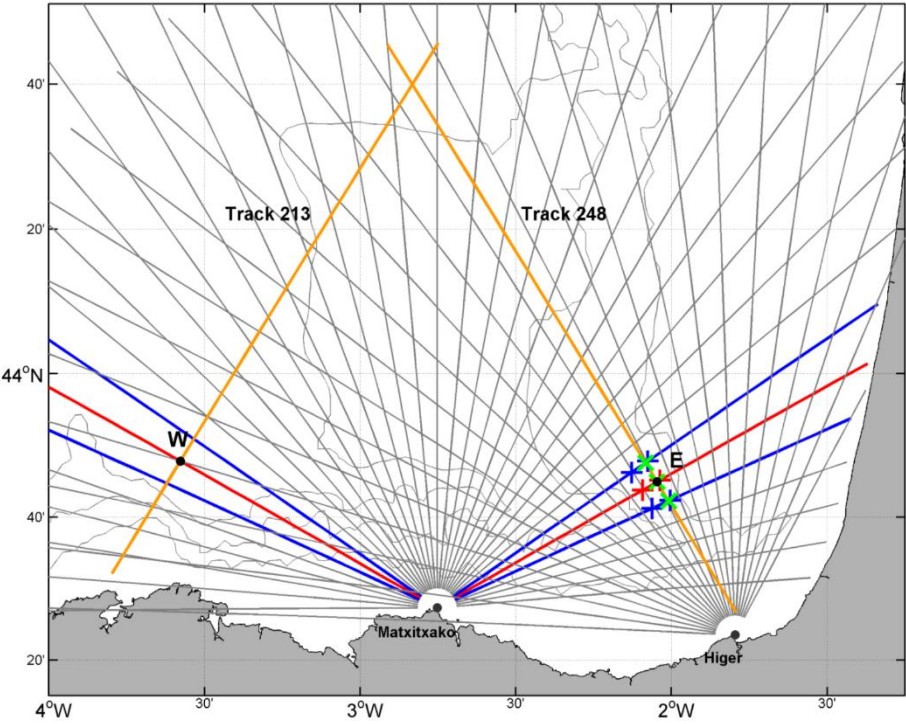

**Figure 2.** Schematic view of the pointwise comparison and of the data used for this approximation. Jason-2's 213 and 248 tracks are depicted by orange lines. The HFR radial directions from both sites are represented by grey lines and the selected radial directions (from Matxitxako site) for the pointwise comparison are plotted in red (i.e. the central radial orthogonal to the track) and in blue (the adjacent radials). Points E and W, are those along both tracks, where each of the HFR radial directions are orthogonal to the Jason-2's tracks directions. At point E, the blue and red crosses show the selected points of HFR radial directions where radial currents are averaged in a first step to obtain the corresponding values on the track (green crosses). Then, the along-track green crosses are averaged to obtain the corresponding HFR current values in the point at issue (the same process is carried out for point W, but it is not represented in the figure). Grey lines: 200, 1000, and 2000 m isobaths.





**Figure 3.** Across-track currents measured by altimetry+Ekman and HFR radials. (a) CTOH, CMEMS and HFR data sets in point $E_R$. (b) CTOH-HFR and CMEMS-HFR residuals in point $E_R$. (c) CTOH, CMEMS and HFR data sets in point $W_R$. (d) CTOH-HFR and CMEMS-HFR residuals in point $W_R$. Black arrows depict the slope current intensifications mentioned in the text.





**Figure 4.** Monthly statistical parameters of the comparison between $AC_G$ ($+AC_E$) and $AC_{HFR,R}$ in the points $E_R$ and $W_R$ for both altimetry products: (a) CTOH $AC_G$ in point $W_R$, (b) CTOH $AC_G$ in point $E_R$, (c) CMEMS $AC_G$ in point $W_R$, (d) CMEMS $AC_G$ in point $E_R$. The normalized RMSD is the quotient between the RMSD and the values of the corresponding $AC_{HFR,R}$.





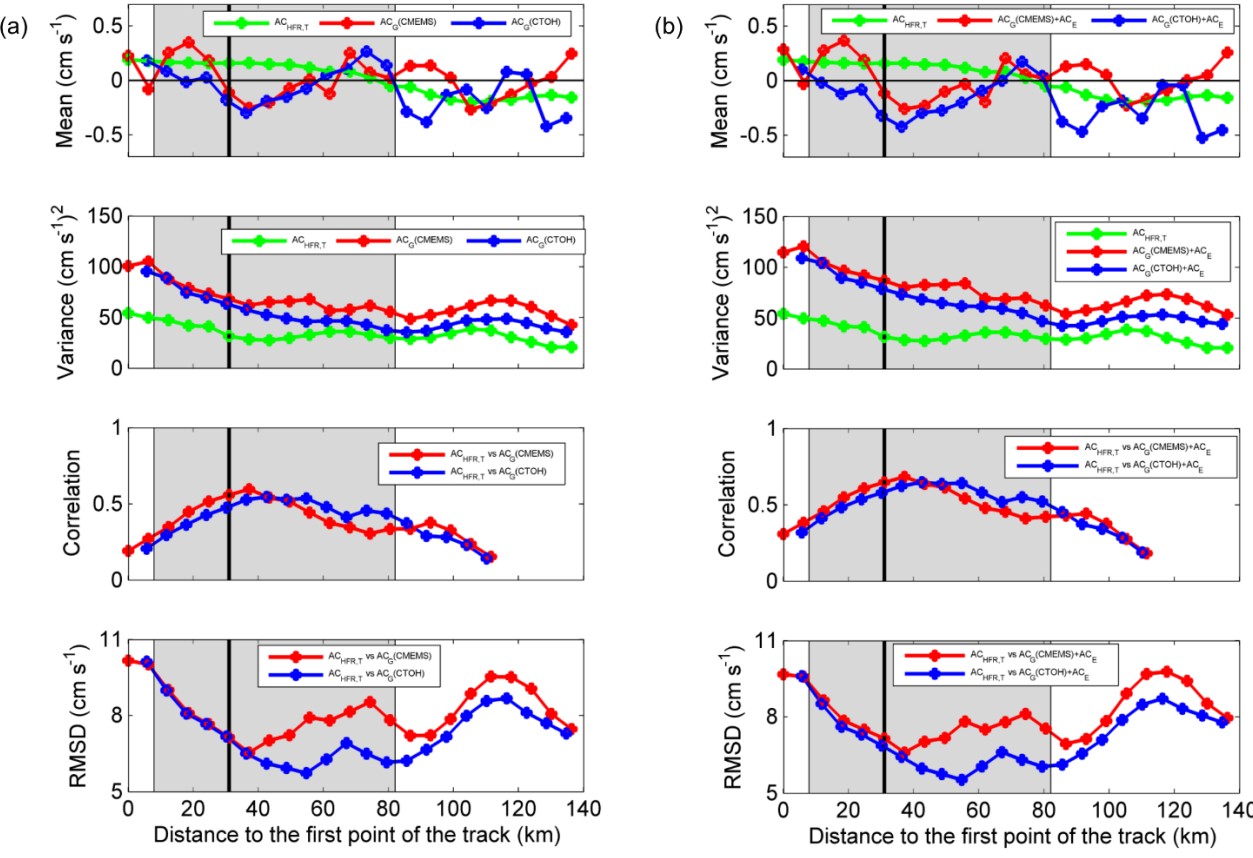

**Figure 5.** Mean and variance along the track 248 of the different data sets ($AC_{HFR,T}$, $AC_G$ (CTOH) and $AC_G$ (CMEMS)) and correlation and RMSD between $AC_G$ and $AC_{HFR,T}$. (a) $AC_{HFR,T}$ vs $AC_G$. (b) $AC_{HFR,T}$ vs $AC_G + AC_E$. The grey coloured area corresponds to the slope between 200 m and 1000 m isobaths. The black line is the location of the point E and the first point of the track is around 11 km away from the coast. Correlation values are not plotted for confidence levels under 90 % (i.e. in the points along the track with a distance from the first point beyond 120 km). Note that the first point is different for CTOH and CMEMS (which are around 10.5 km and 13 km away from the coast respectively) and that the spacing between the points is slightly different. Moreover, the first point in CTOH is removed because it is an outlier. The radar points are the same as CMEMS points.



**Fig. 6.** Time evolution from 1 January 2009 to 1 July 2015 along the track 248 ($y$ axis shows distance to the first point of the track in km) of: (a) CMEMS SLA' (cm), (b) $AC_E$ (cm s⁻¹) and (c) $AC_{HFR}$ (cm s⁻¹). The distance from the first point of the track to the coast is around 6 km. The horizontal black lines delimit the slope area between the 200 and 1000 m isobaths. The black diamonds depict the IPC intensification signals whereas the black inverted triangles show the eddy events, all mentioned in the text.



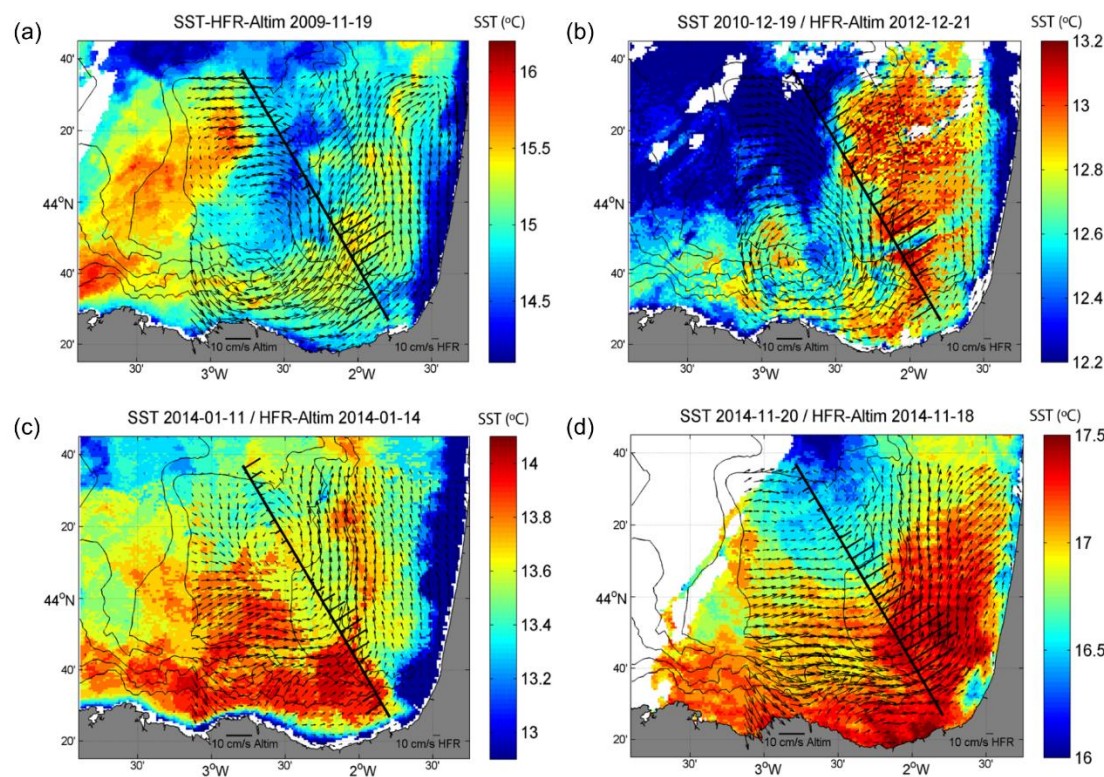

**Figure 7.** Snapshots showing four slope current intensification events observed by HFR, altimetry and SST (see the dates of the events
depicted in Fig. 6) in November 2009 (top left), December 2010-January 2011 (top right), January 2014 (bottom left) and November 2014
(bottom right). The small arrows depict the HFR current fields (not rotated) whereas the thick ones indicate the across-track $AC_G$. The black
line shows the altimeter's track. The colormap depicts the sea surface temperature (ºC) with values referenced in each colour palette. Note
that the dates corresponding to the data are not equal for SST or for HFR and $AC_G$ (specified in each picture's title).





**Figure 8.** Four mesoscale eddies observed in the study area. The dots show the points of the track 248 of the CMEMS database. SLA′ (cm) values are indicated in the colour palette. Black arrows depict the HFR current fields. Red arrows correspond to across-track $AC_G$ derived from the SLA′ values in the dots. Grey lines: 200, 1000, and 2000 m isobaths. Note that the scale of the arrows is not the same in all the subplots.