# Peer review of "Joint analysis of coastal altimetry and HF radar data: observability of seasonal and mesoscale ocean dynamics in the Bay of Biscay"

_Ocean Science, 2018_

## Referee Comment (RC1) · Anonymous Referee #1 · 17 Apr 2018

HF radar and coastal altimetry-derived surface currents are quantitatively compared first at certain locations and then along the altimetry ground tracks in the southeastern Bay of Biscay during multiple years from 2009 to 2015. The two data sets are further used to describe the mesoscale dynamics of the Iberian Poleward Current and eddies in the region. It is demonstrated that a combination of different coastal ocean observing systems may provide a better understanding of the coastal ocean processes. Despite the limitations of the ocean observation techniques, the results are encouraging. Also, the topic is interesting and the paper is informative. I would recommend it be accepted for publication after minor revision.

Specific comments:

Correlation coefficient is used to quantify the comparison between two time series without any statistical significance test. It is important to mention whether a correlation is statistically significant or not, because a high correlation coefficient may not be meaningful at all without a significance test. Also, "correlation coefficients" are simply written as "correlations" in many cases, which should be clearly specified.

Throughout the manuscript, two decimal places are used for ocean current velocities. In my opinion, one decimal digit would be enough. This is because the accuracy of HF radar current measurement is about several cm/s, and the errors in altimetry-derived currents are even larger.

Technical errors:

Page 1, Line 9, "resolution" should be "resolutions".

Page 1, Line 20, "higher" should be changed to "larger".

Page 1, Line 25, "in" should be changed to "by".

Page 2, Lines 1-2, it would be good to refer to a recent book in this topic (Liu et al., 2015).

Page 2, Line 17, "resolution" should be "resolutions".

Page 2, Line 17, "of" should be removed.

Page 5, Line 6, the single sentence paragraph should be combined with the previous paragraph.

Page 5, Line 14, the single sentence paragraph should be combined with the previous paragraph.

Page 5, Line 23, "except" should be changed to "except for".

Page 7, Line 27, what value is used for $C_d$?

Page 9, Line 28, "the higher is the correlation, the smaller is the difference . . ." should be changed to "the higher the correlation, the smaller the difference . . .". Both "is" should be removed from the sentence.

Page 14, Line 27, "Altimetry has also its own limitations" should be changed to "Altimetry also has its own limitations".

Reference: Liu, Y., Kerkering, H., and Weisberg, R.H. (Eds): Coastal Ocean Observing Systems, 461 PP., ISBN 978-0-12-802022-7, Elsevier (Academic Press), London, UK, 2015.

---

## Referee Comment (RC2) · Anonymous Referee #2 · 30 Apr 2018

the paper focuses on the validation of two different altimetry products and aims at demonstrating the complementary with high-frequency radar observations of surface currents in the bay of Biscay. the manuscript has a few flaws, that should be accounted for properly before it is accepted for publication. details are given below for the Authors and the Editorial Board.

Abstract: - define "surface currents"; - correct the spatial and temporal resolution: depending on the HFR system, they can be as low as 300m to 12km, 5 min to 3 hours; same corrections apply within the text - line 20: what variability is the Author referring to? why is it so? it would be useful to have these details in the abstract - line 25: is this

[Figure]

correlation increase statistically significant? is there a real benefit in including a simplified Ekman current model to the data, given the amount of processing the dataset already go through? - check grammar and break sentences to improve readability (mainly within the manuscript) - lines 20-25: I personally would rephrase this sentence in order to focus on benefits first and limitations after; for instance, something like: "Both HFR and altimetry capture the main oceanographic features in the region (the IPC and the mesoscale eddies), however performances reduce in the areas closer to shore because of ....", or similar.

Introduction: - page2, line4: references to Jerico and Jerico-Next should be added; - page2, line6: cit: "...best possible quality indicators..." of what? - page2, line19: definition of "HF" is missing; the guess is, it means High-Frequency spectral components, but it confuses with the acronym HFR; - page2, line20: HFR do not measure "inertial waves" but can resolve "inertial currents" if the proper grid resolution is set up; - page3, line 12: missing network after HFR - page3, line 15-18: one of the major motivations of this present work - that is, the comparison of the two products - is not stressed out properly in my opinion; this is actually te added value to this manuscript.

Section 2.1.1: This section should be rephrased and detailed more because as it is now it contains a series of significant errors. - radial velocities are not measured directly; they are derived from the inversion of the 1st order scatter from Bragg-matching waves; - operational range is usually frequency and bandwidth dependent; low-frequency systems have usually narrower bandwidths thus boosting range but with inversely-proportional range resolution - 40KHz bandwidth should provide a radial range resolution of ∼3.7km, not 5km as stated; - "noise to signal ratio" should be the opposite: signal to noise ratio; - is it the RT HFR product being used, or the reprocessed DM data set used instead? - "receipt antenna pattern" should be "receive antenna pattern"; - bi-annual calibration: performed every two years, or twice per year? - page 4, lines 5-13: this section is confusing. Gurgel (1994) and Lipa and Barrick (1983) proposed the unweighted least-square fit for the WERA and the Codar systems. the

first does a 1-1 match of radials from two stations, the second uses a spatial search radius. The OMA analysis has nothing to do with this. OMA was developed by Kaplan and Lekien (2007). My understanding is the following: radials in polar coordinate systems from the two separate stations were mapped to currents on a cartesian grid using the HFR_Progs Matlab package; then, an OMA analysis was performed for gap-filling purposes. Since the results of the conventional least-squares approach were similar to the OMA output, it was decided to use the OMA products for the following analysis. If that is the case, there is at least one motivation for me to ask 1, if there is any quantitative comparison between the OMA and the LS fit with any other data set (see for instance Cosoli et al., 2015, who tested the EOF interpolation versus the conventional LS fit in the Malta Channel); 2, to have at least a map of the comparison metrics between the OMA - LS fit products. The reason being: OMA is fitting a limited number of modes which will inevitably loose some observed structures, and most likely adds some spurious structure that needs to be accounted for properly.

Section 2.1.2 This section also should be checked thoroughly and detailed more. - page4, lines28-32: details should be provided about this data processing approach, especially in relation to the spatial filtering approach. References should be added to the Loess filter because it needs to be understood properly in order to avoid biases at ranges from the coastline within the filter spatial cut-off length. if the filtered products are used to derive the along-track geostrophic currents, I would expect a systematic bias between HFR and satellite in the coastal regions; this would explain for instance the biases documented in Figure 5 for the first 4-5 bins; also, it would most likely explain why correlation is maximised between km 40 and 50 (3rd panels, figure 5 a and b) - page5, lines1:5: same considerations as above apply to this dataset

Section 2.2 - page5, lines17-24: while the moving-average filter is probably fine in removing the low-frequency components from the HFR data set, it would be useful to have also some quantitative results of the sensitivity study about the 2, 5, 10, 15 d windows. How was the phase shift introduced by the MA process handled, for instance?
Given the spatial smoothing the altimetry data goes through, I believe a similar thing should be done for the HFR data set, so that to avoid any processing bias. - page 6: I there is something wrong with eqtn. 3; this applies to a standard orthogonal Cartesian x-y plane with x axis pointing eastwards, y axis pointing northwards and z axis ponting to the opposite direction of gravity; not clear in the text if the geostrofic velocities are computed in this coordinate system. assuming it is so, however, the derivative should be computed along y if one wants the across-track velocity, not x: u=-g/f*DSLA/Dy; -page 6, lines18-31: more details are needed in regard to this. I assume that the comparison is performed after projecting the geostrofic currents in the direction of the radar stations, so to have a "true" comparison between the radial currents. That would be fine if the radars was error-free, which is not the case. Usually, the direction-finding radars suffer from systematic and unpredictable errors in the determination of the incoming signal, which results in statistically significant bearing offsets (see Emery et al., 2004, for additional details). I think this analysis should be extended to a few more angular sectors or the potential limitations properly acknowledged in the text - page 7, eqtn. 5: the bulk-flux formula described here has no references-it should be added; is the stress computed at the standard 10-m height? what formulation is used for the drag coefficient? is it wind speed dependent or independent? - page 8, lines 7-8: HF again I suspect stays for "high-frequency"; so, the Ekman currents are computed then low-pass filtered with the same 10-d moving average filter. same considerations as before apply also to this product - a spatial filtering should also be applied

Section 3 - I would like to see the actual 95%-99% CL to correlation and statistics; are changes in correlation statictically significant? based on Table1, the high standard deviations do compensate for any changes in mean values, and as such I would be cautious in interpreting similar variations - it is stated that in general adding the Ekman currents decreases rmsd but adds variability; it would be interesting to see a plot of these terms and try understand if the added variablity reflects in the intrinsic variability of the Ekman term

- Figure 5 needsome additional analysis and comments: interestingly, HFR-altimetry correlation is maximised at around 40-60 km which is comparable to the size of the altimetry spatial filtering widow; the HFR dataset shows an inversion at the edge of the grey-marked area (which corrsponds to the 1000m isobath); but neither the CMEMS or the CTOH products follow that pattern. why is it so? what are the sources of a similar disagreement?

- While in general there is an agreement between the mesoscale patterns (Figure 8 for insytance), comparison is poor in the region close to shoreline where the altimetry products are often in oppposite direction to the HFR data. In this sense, it would be interesting: 1, to investigate a bit further the assumptions of geostrophic balance in the boundary regions; 2, try to merge the altimetry and HFR data so to correct and in this way maximise the two products

---

## Author Comment (AC1) · 22 Jun 2018

Our comments to both reviewers and the marked version of the modified manuscript were uploaded in the form of a supplement zip file.

Please also note the supplement to this comment:
https://www.ocean-sci-discuss.net/os-2018-33/os-2018-33-AC1-supplement.zip

---

## Author Response (AR1)

**AUTHOR'S RESPONSE**

Note that all the locations of the changes mentioned herein after correspond to the locations in the "clean" manuscript, not the marked-up version.
* * *
**Anonymous Referee #1**

HF radar and coastal altimetry-derived surface currents are quantitatively compared first at certain locations and then along the altimetry ground tracks in the southeastern Bay of Biscay during multiple years from 2009 to 2015. The two data sets are further used to describe the mesoscale dynamics of the Iberian Poleward Current and eddies in the region. It is demonstrated that a combination of different coastal ocean observing systems may provide a better understanding of the coastal ocean processes. Despite the limitations of the ocean observation techniques, the results are encouraging. Also, the topic is interesting and the paper is informative. I would recommend it be accepted for publication after minor revision.
* * *
**Dear reviewer,**

**First of all, thank you for showing the technical errors. They really help to improve the manuscript and we have addressed all of the into the text. Thank you for your specific comments as well. We did not realize that we had not mentioned the confidence level of the correlations, which is 90%. We have also included it as explained below.**

**Best regards,**
**Ivan Manso**

Comments are enumerated

AR = author's response

AC = author's changes in manuscript

1. Correlation coefficient is used to quantify the comparison between two time series without any statistical significance test. It is important to mention whether a correlation is statistically significant or not, because a high correlation coefficient may not be meaningful at all without a significance test.

AR: We considered the significance (90%), therefore, all the results shown in the manuscript are significant. However, it was not mentioned in the text. Thank you for this comment.

AC: We have added this information at page 8, line 15

2. "correlation coefficients" are simply written as "correlations" in many cases, which should be clearly specified.

AR: We have specified it.

AC: Page 8, line 14.

3. Throughout the manuscript, two decimal places are used for ocean current velocities. In my opinion, one decimal digit would be enough. This is because the accuracy of HF radar current measurement is about several cm/s, and the errors in altimetry-derived currents are even larger.

AR: We have changed the values, using only one decimal place throughout all the manuscript.

4. Page 1, Line 9, "resolution" should be "resolutions".

AR: Done.

AC: Page 1, line 9.

5. Page 1, Line 20, "higher" should be changed to "larger".

AR: The sentence has been rephrased and therefore this adjective has been removed.

6. Page 1, Line 25, "in" should be changed to "by".

AR: Done.

AC: Page 1, line 24.

7. Page 2, Lines 1-2, it would be good to refer to a recent book in this topic (Liu et al., 2015).

AR: Done.

AC: Page 2, line 1.

8. Page 2, Line 17, "resolution" should be "resolutions".

AR: Done.

AC: Page 2, line 17.

9. Page 2, Line 17, "of" should be removed.

AR: Sorry, but we have not found any 'of' in that point.

10. Page 5, Line 6, the single sentence paragraph should be combined with the previous paragraph.

AR: We have rephrased and moved this sentence to page 4, line 30.

11. Page 5, Line 14, the single sentence paragraph should be combined with the previous paragraph.

AR: Done.
AC: Page 5, Line 18.

12. Page 5, Line 23, "except" should be changed to "except for".

AR: Done.
AC: Page 5, line 27.

13. Page 7, Line 27, what value is used for Cd?

AR: The value we used was the one proposed by Large and Pond (1981) which depends on the velocity of the wind ($U_{10}$):

$$C_D = 1.2 \text{ x } 10^{-3} \qquad\qquad \text{for} \quad 4 < U_{10} < 11 \text{ m s}^{-1}$$
$$C_D = 10^{-3} (0.49 + 0.065 \, U_{10}) \quad \text{for} \quad 11 < U_{10} < 25 \text{ m s}^{-1}$$

Note that we have considered $1.2 \text{ x } 10^{-3}$ for every value smaller than 11 m s$^{-1}$ and $10^{-3}$ (0.49 + 0.065 $U_{10}$) for larger values.

AC: We have added the reference to Large and Pond (1981) at page 8, line 1.

14. Page 9, Line 28, "the higher is the correlation, the smaller is the difference…" should be changed to "the higher the correlation, the smaller the difference…". Both "is" should be removed from the sentence.

AR: Done.
AC: Page 9, line 29.

15. Page 14, Line 27, "Altimetry has also its own limitations" should be changed to Altimetry also has its own limitations".

AR: Done.

AC: Page 14, line 30.
* * ** * *
**Anonymous Referee #2**

the paper focuses on the validation of two different altimetry products and aims at demonstrating the complementary with high-frequency radar observations of surface currents in the bay of Biscay. the manuscript has a few flaws, that should be accounted for properly before it is accepted for publication. details are given below for the Authors and the Editorial Board.
* * *
**Dear reviewer,**

**Thank you for your thorough reading of the text and the exhaustive review. Your fruitful comments have improved the article.**

**Best regards,**
**Ivan Manso**

Comments are enumerated
AR = author's response
AC = author's changes in manuscript

Abstract:

1. define "surface currents"

AR: Done.

AC: Page 1, Line 8.

2. correct the spatial and temporal resolution: depending on the HFR system, they can be as low as 300m to 12km, 5 min to 3 hours; same corrections apply within the text.

AR: Done.

AC: Page 1, Line 9.

3.  line 20: what variability is the Author referring to? why is it so? it would be useful to have these details in the abstract

AR: The line has been rephrased (see comment 6) and this variability it is not mentioned any more.

4.  line 25: is this correlation increase statistically significant? is there a real benefit in including a simplified Ekman current model to the data, given the amount of processing the dataset already go through?

AR: Thank you for this comment, the significance of the correlation is 90%, but we forgot to mention it in the text, therefore, it has been added at page 8, line 15. With regard to the addition of the Ekman current model, it is true that the qualitative benefit is not high, but apart from providing a moderate benefit it is methodologically interesting and gives way to a better analysis.

5.  check grammar and break sentences to improve readability (mainly within the manuscript)

AR: done.

6.  lines 20-25: I personally would rephrase this sentence in order to focus on benefits first and limitations after; for instance, something like: "Both HFR and altimetry capture the main oceanographic features in the region (the IPC and the mesoscale eddies), however performances reduce in the areas closer to shore because of ....", or similar.

AR: Rephrased.

AC: Page 1, Lines 20-24.

Introduction:

7.  page2, line4: references to Jerico and Jerico-Next should be added

AR: Done.

AC: Page 2, Line 2.

8.  page2, line6: cit: "...best possible quality indicators..." of what?

AR: This sentence has been rephrased.

AC: Page 2, Line 6.

9.  page2, line19: definition of "HF" is missing; the guess is, it means High-Frequency spectral components, but it confuses with the acronym HFR

AR: Changed to high-frequency.

AC: Page 2, Line 19.

10. page2, line20: HFR do not measure "inertial waves" but can resolve "inertial currents" if the proper grid resolution is set up

AR: Thank you, it was a mistake. It has been corrected.

AC: Page 2, Line 20.

11. page3, line 12: missing network after HFR

AR: Done.

AC: Page 3, Line 11.

12. page3, line 15-18: one of the major motivations of this present work - that is, the comparison of the two products - is not stressed out properly in my opinion; this is actually te added value to this manuscript.

AR: Rephrased.

AC: Page 2, Lines 17-19.

Section 2.1.1

13. radial velocities are not measured directly; they are derived from the inversion of the 1st order scatter from Bragg-matching waves

AR: We have changed 'measure' to 'infer'.

AC: Page 3, Line 24.

14. operational range is usually frequency and bandwidth dependent; low-frequency systems have usually narrower bandwidths thus boosting range but with inversely-proportional range resolution - 40KHz bandwidth should provide a radial range resolution of _3.7km, not 5km as stated.

AR: It is true; however, the range cell and angular resolutions are set to 5 km and 5° respectively.

AC: We have added this clarification in page 3, line 31.

15. "noise to signal ratio" should be the opposite: signal to noise ratio.

AR: Done.

AC: Page 4, Line 1.

16. is it the RT HFR product being used, or the reprocessed DM data set used instead?

AR: The DM one.

17. "receipt antenna pattern" should be "receive antenna pattern"

AR: Done.

AC: Page 4, Line 2.

18. bi-annual calibration: performed every two years, or twice per year?

AR: It has been performed at least every two years.

AC: We have clarified it at page 4, line 2.

19. Page 4, lines 5-13: this section is confusing. Gurgel (1994) and Lipa and Barrick (1983) proposed the unweighted least-square fit for the WERA and the Codar systems. The first does a 1-1 match of radials from two stations, the second uses a spatial search radius. The OMA analysis has nothing to do with this. OMA was developed by Kaplan and Lekien (2007). My understanding is the following: radials in polar coordinate systems from the two separate stations were mapped to currents on a cartesian grid using the HFR_Progs Matlab package; then, an OMA analysis was performed for gap-filling purposes. Since the results of the conventional least-squares approach were similar to the OMA output, it was decided to use the OMA products for the following analysis. If that is the case, there is at least one motivation for me to ask 1, if there is any quantitative comparison between the OMA and the LS fit with any other data set (see for instance Cosoli et al., 2015, who tested the EOF interpolation versus the conventional LS fit in the Malta Channel); 2, to have at least a map of the comparison metrics between the OMA - LS fit products. The reason being: OMA is fitting a limited number of modes which will inevitably loose some observed structures, and most likely adds some spurious structure that needs to be accounted for properly.

AR: Thank you for the in-depth revision of this part. The whole paragraph has been checked and rewritten to allow more clear explanations. Indeed, this is not the first time we used both LS fit and OMA for total current retrieval. Quantitative comparison between both methods have been performed in Solabarrieta et al. 2016, from a Lagrangian perspective and in Hernandez-Carrasco (this same issue- see https://www.ocean-sci-discuss.net/os-2018-26/).

(Solabarrieta, L., Frolov, S., Cook, M., Paduan, J., Rubio, A., González, M., Mader, J., and Charria, G.: Skill Assessment of HF Radar-Derived Products for Lagrangian Simulations in the Bay of Biscay, Journal of Atmospheric and Oceanic Technology, 33, 2585–2597, https://doi.org/10.1175/JTECH-D-16-0045.1, 2016.)

AC: page 4, lines 5-12.

Section 2.1.2

20. page4, lines28-32: details should be provided about this data processing approach, especially in relation to the spatial filtering approach. References should be added to the Loess filter because it needs to be understood properly in order to avoid biases at ranges from the coastline within the filter spatial cut-off length. if the filtered products are used to derive the along-track geostrophic currents, I would expect a systematic bias between HFR and satellite in the coastal regions; this would explain for instance the biases documented in Figure 5 for the first 4-5 bins; also, it would most likely explain why correlation is maximised between km 40 and 50 (3rd panels, figure 5 a and b) - page5, lines1:5: same considerations as above apply to this dataset.

AR: We have bias if we have systematic errors in SLA at the coast which are not filtered by the data editing process and then extrapolated offshore by the filter. This is true for both CTOH and CMEMS products even if the filters and cut off frequency (and then area where errors are extrapolated) are not the same. And indeed, we know that the accuracy of altimetry data is lower in the 20-30 km coastal band, so it might be a source of differences between altimetry and HFR data. But the larger the oceanic signal (larger signal-over-noise ratio), the lower this effect will be observed anyway. Additionally, note that SLA errors approaching the coast are not usually systematic as they are mainly due to several sources (signal perturbations in altimeter/radiometer footprint, SSB, tidal & DAC corrections) which vary from one cycle to another.

AC: An explanation with regard to this has been added in page 5, lines 8-12. The reference to the Loess filter has been added in page 5 line 3 (Cleveland, W.S. and Devlin, S.J.: Locally Weighted Regression: An Approach to Regression Analysis by Local Fitting, JASA, 83, 596-610,1988).

Section 2.2

21. page5, lines17-24: while the moving-average filter is probably fine in removing the low-frequency components from the HFR data set, it would be useful to have also some quantitative results of the sensitivity study about the 2, 5, 10, 15 d windows. How was the phase shift introduced by the MA process handled, for instance?

AR: These are the results of the sensitivity tests carried out for the pointwise comparison, in terms of correlation, where the 10-d window seems to be the most reasonable choice:

| CTOH | CMEMS |
| --- | --- |

| | | | |
|---|---|---|---|
| Point E: | Point W: | Point E: | Point W: |
| 2 days → r=0.37 | 2 days → r=0.59 | 2 days → r=0.45 | 2 days → r=0.53 |
| 5 days → r=0.42 | 5 days → r=0.60 | 5 days → r=0.51 | 5 days → r=0.60 |
| 10 days → r=0.48 | 10 days → r=0.64 | 10 days → r=0.53 | 10 days → r=0.60 |
| 15 days → r=0.50 | 15 days → r=0.59 | 15 days → r=0.54 | 15 days → r=0.54 |

With regard to the phase shift, there is not such an effect because the data used in the MA is always the original. That is, the filtered data is not used in the MA. Therefore, it is independent of doing it forward or backward.

22. Given the spatial smoothing the altimetry data goes through, I believe a similar thing should be done for the HFR data set, so that to avoid any processing bias.

AR: Although the radar data was already smoother than the altimetry one, we have applied two spatial filters to the radar data for the along-track comparison in order to see which the effects of this filtering are. In the cases where the Ekman currents are used, they are also spatially filtered. In the next table the correlations between altimetry and HFR are shown at the two points related to this comparison. The no cutoff results are the same results of the Table 1 (in the text), however more decimal numbers are used to show which are the effects of the filtering. The correlations scarcely change and in some cases are even lower than the original ones, therefore the spatial filtering does not improve the analysis. As it can be seen in the figures below, the results do not change neither along the track. The mean values and the std of the radar are a bit lower as we use bigger cutoffs, however, as it has been said the difference is almost imperceptible. Therefore, we are not going to make any change in the analysis in this sense.

| Cutoff | r $(AC_G)$ | | r $(AC_G + AC_E)$ | |
|---|---|---|---|---|
| | $E_T$ | $Max_T$ | $E_T$ | $Max_T$ |
| 65 km | CTOH: 0.5210 | CTOH: 0.5317 | CTOH: 0.6249 | CTOH: 0.6447 |
| | CMEMS: 0.5476 | CMEMS: 0.5783 | CMEMS: 0.6339 | CMEMS: 0.6694 |
| 40 km | CTOH: 0.5294 | CTOH: 0.5421 | CTOH: 0.6284 | CTOH:0.6503 |
| | CMEMS: 0.56 | CMEMS: 0.5927 | CMEMS: 0.6447 | CMEMS: 0.6794 |
| No cutoff (values of the table) | CTOH: 0.5261 | CTOH: 0.5452 | CTOH: 0.6223 | CTOH: 0.6472 |
| | CMEMS: 0.5572 | CMEMS: 0.5951 | CMEMS: 0.6462 | CMEMS: 0.6824 |

23. page 6: I there is something wrong with eqtn. 3; this applies to a standard orthogonal Cartesian x-y plane with x axis pointing eastwards, y axis pointing northwards and z axis ponting to the opposite direction of gravity; not clear in the text if the geostrofic velocities are computed in this coordinate system. assuming it is so, however, the derivative should be computed along y if one wants the across-track velocity, not x: u=-g/f*DSLA/Dy.

AR: It is mentioned in page 6 line 12 that x is the along-track distance. Therefore, we are in a rotated plane with respect to the usual coordinate system (where x axis points eastwards and y axis points northwards). For both altimetry tracks (213 and 248) the y axis has a positive westward component and since we assume eastward direction as the positive one (it is the direction of the main currents in the area) we add the minus symbol (considering the -y direction as the positive one) to achieve this.

24. page 6, lines18-31: more details are needed in regard to this. I assume that the comparison is performed after projecting the geostrofic currents in the direction of the radar stations, so to have a "true" comparison between the radial currents. That would be fine if the radars was error-free, which is not the case. Usually, the direction-finding radars suffer from systematic and unpredictable errors in the determination of the incoming signal, which results in statistically significant bearing offsets (see Emery et al., 2004, for additional details). I think this analysis should be extended to a few more angular sectors or the potential limitations properly acknowledged in the text.

AR: The point of this method is to compare currents in across-track direction. If we extend our analysis to a few more angular sectors we would be adding currents that are not in such direction. Actually, as mentioned in the text, 3 radials are already considered to compute the across-track currents in each point (E and W). Another possibility would be to use additional altimeter tracks but the only one in the HFR footprint area in addition to that of Jason altimeter would be that of Sentinel, so

it would be difficult to evaluate if the different correlation obtained in both tracks would be due to offsets in the bearing or to differences among SLA data from these two altimeter products.

25. page 7, eqtn. 5: the bulk-flux formula described here has no references-it should be added; is the stress computed at the standard 10-m height? what formulation is used for the drag coefficient? is it wind speed dependent or independent?

AR: With regard to the drag coefficient, the value we used was the one proposed by large and Pond (1981) which depends on the velocity of the wind at the standard 10-m height ($U_{10}$):

$$C_D = 1.2 \times 10^{-3} \qquad \text{for} \quad 4 < U_{10} < 11 \text{ m s}^{-1}$$
$$C_D = 10^{-3} (0.49 + 0.065\ U_{10}) \quad \text{for} \quad 11 < U_{10} < 25 \text{ m s}^{-1}$$

Note that we have considered $1.2 \times 10^{-3}$ for every value smaller than 11 m s$^{-1}$ and $10^{-3}$ (0.49 + 0.065 $U_{10}$) for larger values.

AC: We have added Large and Pond (1981) reference at page 8, line 1. A reference (Stewart, R.H.: Introduction to Physical Oceanography, Texas A & M University, 2004) has been added regarding the bulk-flux formula (eq. 5) at page 7, line 28.

26. page 8, lines 7-8: HF again I suspect stays for "high-frequency"; so, the Ekman currents are computed then low-pass filtered with the same 10-d moving average filter. same considerations as before apply also to this product - a spatial filtering should also be applied.

AR: Same answer of the comment 22.

Section 3

27. I would like to see the actual 95%-99% CL to correlation and statistics; are changes in correlation statictically significant?

AR: Same answer of comment 4.

28. based on Table1, the high standard deviations do compensate for any changes in mean values, and as such I would be cautious in interpreting similar variations.

AR: You are right.

AC: A clarification has been added at page 11, line 13.

29. it is stated that in general adding the Ekman currents decreases rmsd but adds variability; it would be interesting to see a plot of these terms and try understand if the added variablity reflects in the intrinsic variability of the Ekman term.

AR: The Ekman component is in good agreement with HFR and altimetry series, thus when it is added to the altimetry it strengthens it effects, and consequently increases the variability. See next figures:

**Ekman current addition: in E**

[Figure]

**Ekman current addition: in W**

[Figure]

30. Figure 5 needsome additional analysis and comments: interestingly, HFR-altimetry correlation is maximised at around 40-60 km which is comparable to the size of the altimetry spatial filtering widow; the HFR dataset shows an inversion at the edge of the grey-marked area (which corrsponds to the 1000m isobath); but neither the CMEMS or the CTOH products follow that pattern. why is it so? what are the sources of a similar disagreement?

AR: Although the correlation is maximized at around 40-60 km it already starts decreasing before the 1000 m isobath is reached. In addition, when the analysis with the spatially filtered radar data was carried out (see figures at comment 22), the pattern of the correlation did not change.

With regard to the inversion at the edge of the slope, it is mentioned in the text that the mean values of the currents are really small and that they are very close to zero. Therefore, the differences that are shown in figure 5 are subtle. Additionally, this greater variability of the altimetry data agrees with what it has been mentioned and shown throughout the manuscript: a larger variability in the altimetry data set. The tendency that the HFR data set shows is expectable (westward currents in winter and eastward currents in summer) and it agrees with the altimetry data if we carry out a seasonal analysis (see figures below). It can be observed that the mean value is going to be close to zero, and if we consider the spring and autumn data, we finally obtain the results of Figure 5.

[Figure]

31. While in general there is an agreement between the mesoscale patterns (Figure 8 for insytance), comparison is poor in the region close to shoreline where the altimetry products are often in oppposite direction to the HFR data. In this sense, it would be interesting: 1, to investigate a bit further the assumptions of geostrophic balance in the boundary regions; 2, try to merge the altimetry and HFR data so to correct and in this way maximise the two products

AR: Thank you for these comments. Since they are interesting work lines for the future, we have added them in the discussion.

AC: Page 15, Lines 2-4.

**List of relevant changes:**

In general, there are not any major changes in the manuscript, since there are not any changes in the analyses made. Most of the changes are improvements in the writing. These are the most relevant ones:

- Page 1, Line 8: Definition of the surface currents.
- Page 1, Line 9: Specification of HFR resolution.
- Page 1, Lines 20-24: Rephrased sentence.
- Page 2, Line 1: Added reference (with its corresponding change in the reference section).
- Page 3, Line 31: Rephrased sentence.
- Page 4, Lines 4-7: Rephrased sentence.
- Page 5, Lines 8-12: An explanation about the sources of differences between radar and altimetry near the coast has been added.
- Page 7, Line 28: Added reference (with its corresponding change in the reference section).
- Page 8, Line 1: Added reference (with its corresponding change in the reference section).
- Page 8, Line 15: Specification of the minimum confidence levels considered for the correlations.
- Page 11, Line 12-13: Explanation that the high std values compensate changes in the mean values.
- Page 15, Lines 2-4: Addition of a future work line proposed by the reviewer 2.
- Page 21, Line 10: Explanation for the blank space in Table 1 in $Max_T$ point.
- Throughout the manuscript: The use of one decimal for currents and statistical results as proposed by reviewer 1.

**Marked-up version of the manuscript:**

[revised manuscript text omitted]

---

## Author Response (AR2)

**AUTHOR'S RESPONSE**

**Anonymous Referee #2**

5  Dear reviewer,

Thank you for your final comments. We have taken them into account.

Best regards,

Ivan Manso

10  - Comment 1: p4, l30: max operating range is set to 100km; whereas p4, l6 states 6km;

- Answer 1: We do not see exactly to what the reviewer is referring to. We've double checked the whole paper in search on any other inconsistence on this, without finding anything.

- Comment 2: p4, l30: resolution is set to 1-5km; whereas abstract states ~300m to 10km; this should be checked for
15  consistency.

- Answer 2: We have changed the resolution to 300 m-10 km

- Comment 3: p5, l think the Authors are confusing the radial range resolution with the vector grid resolution. Radar radial resolution is set by bandwidth - the rule-of-thumb for SeaSonde HFR systems as the ones used in this region is
20  radial_resolution_km = 150/BW. so, at 40kHz, the radial range resolution is indeed 3.75km.

- Answer 3: The radial range resolution is 5.12 km. This value is, as pointed by the reviewer, lower than the theoretical range resolution that could be expected for a 40kHZ bandwidth system. The reason is that we are only using 30kHz for operation to prevent potential interferences to other systems. So, in our case both radial range resolution and vector grid resolution are of ~5km. We have corrected the bandwidth data to 30 kHz.

**Topic Editor:**

Dear Stefania,

Thank you for your comment.
30  Best regards,

Ivan Manso

- Comment 1: Page 16, Dussurget, R., Birol, F., Morrow, R., and De Mey, P.: Fine resolution altimetry data for a regional application in the Bay of Biscay, Mar. Geod., 34, 447-476, 2001. The year is 2011 instead.
- Answer 1: done.

5      **List of relevant changes:**

The pages and lines correspond to the manuscript that it is below. Most of the changes are improvements in the writing. These are the most relevant changes, which are related to the comments of the reviewer and the topic editor:
- Page 3, Line 26: The resolution of the radar has been changed to 300 m-10 km, as proposed by the reviewer.
10     - Page 3, Line 28: We have corrected the bandwidth data to 30 kHz.
- Page 16, Line 126: The year of the reference has been changed.

**Marked-up version of the manuscript:**

[revised manuscript text omitted]